# Universal Vision-Language Dense Retrieval: Learning A Unified Representation Space for Multi-Modal Retrieval

**Zhenghao Liu**[1]     **Chenyan Xiong**[2]     **Yuanhuiyi Lv**[1]     **Zhiyuan Liu**[3]     **Ge Yu**[1]

[1]Department of Computer Science and Technology, Northeastern University, Shenyang, China
[2]Microsoft Research, Redmond, USA
[3]Department of Computer Science and Technology, Tsinghua University, Beijing, China
Institute for Artificial Intelligence, Tsinghua University, Beijing, China
State Key Lab on Intelligent Technology and Systems, Tsinghua University, Beijing, China

## Abstract

This paper presents Universal Vision-Language Dense Retrieval (UniVL-DR), which builds a unified model for multi-modal retrieval. UniVL-DR encodes queries and multi-modality resources in an embedding space for searching candidates from different modalities. To learn a unified embedding space for multi-modal retrieval, UniVL-DR proposes two techniques: 1) Universal embedding optimization strategy, which contrastively optimizes the embedding space using the modality-balanced hard negatives; 2) Image verbalization method, which bridges the modality gap between images and texts in the raw data space. UniVL-DR achieves the state-of-the-art on the multi-modal open-domain question answering benchmark, WebQA, and outperforms all retrieval models on the two subtasks, text-text retrieval and text-image retrieval. It demonstrates that universal multi-modal search is feasible to replace the divide-and-conquer pipeline with a united model and also benefits single/cross modality tasks. All source codes of this work are available at `https://github.com/OpenMatch/UniVL-DR`.

## 1 Introduction

Although search engines primarily focus on textual data (Singhal et al., 2001), multi-media is necessary to satisfy user needs during retrieval. A user query can be answered by the information in variant formats, such as a text document, or a picture. The growth of multi-media content has been one of the most notable trends on the internet (Mei et al., 2014), and various studies have proved that users prefer more vivid multi-media content in search results (Datta et al., 2008).

Current multi-media search systems often employ a divide-and-conquer approach. As shown in Figure 1(a), they first conduct search in individual modalities, including text, image, video, etc. (Bajaj et al., 2016; Grubinger et al., 2008; Kwiatkowski et al., 2019; Awad et al., 2021), and then fuse the retrieval results from various verticals together, e.g., building another ranking layer on top of these single/cross modality retrievers (Escalante et al., 2008; Grubinger et al., 2008). Both relevance modeling and retrieval result fusion are usually entwined to achieve more accurate multi-modal retrieval results. However, due to the modality gap, they can be only pipeline-modeled in divide-and-conquer, making it challenging to fuse retrieval results from different modalities.

In this paper, we explore the potential of universal multi-modal retrieval to build an end-to-end model and retrieve multi-modality documents for user queries. Illustrated in Figure 1(b), universal multi-modal retrieval maps queries and multi-modality resources to one universal embedding space and retrieves multi-modality candidates via KNN search. As a result, the relevance modeling, cross-modality matching, and retrieval result fusion are done by one model.

More specifically, we propose a Universal Vision-Language Dense Retrieval (UniVL-DR) model to get the representations of queries, texts, and images and learn a tailored vision-language embedding space for multi-modal retrieval. UniVL-DR optimizes the vision-language embedding space using hard negatives (Xiong et al., 2021a) and balances the modalities of these negatives to alleviate

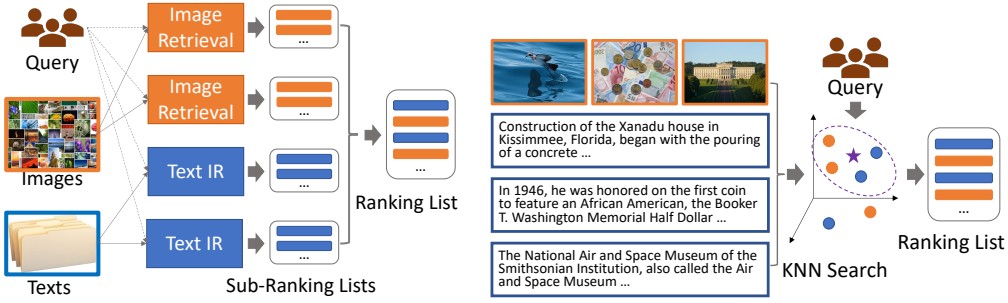

(a) Divide-and-Conquer Multi-Media Search.          (b) Universal Vision-Language Search.

Figure 1: Different Architectures of Multi-Modal Retrieval Systems.

the modality preference of multi-modal retrievers. Furthermore, UniVL-DR introduces an image verbalization method, which regards language as a kind of mentalese (Cavanagh, 2021) and mitigates the modality gap between images and texts. Our image verbalization method first aligns the semantics of image captions and figure pixels (Huang et al., 2021a), and then paraphrases the image facts. It helps to bridge language and vision understanding modules of UniVL-DR via natural language.

To build a multi-modal retrieval benchmark, we leverage a multi-modal question answering (QA) benchmark WebQA (Chang et al., 2022) and convert it to a standard open-domain setting: retrieving multi-modality candidates from text and image collections for a user query. Divide-and-conquer is an intuitive way to build a multi-modal retrieval system and we pre-route queries to oracle modality to show the upper bound performance of such a system. Compared with the divide-and-conquer system, UniVL-DR addresses the retrieval result fusion challenge, achieves state-of-the-art multi-modal retrieval performance, and brings more than 5% improvement in single/cross modality retrieval.

Our experiments show that UniVL-DR learns an effective embedding space for multi-modal retrieval by separating texts and images into different areas and guiding queries to return candidates from corresponding modalities. Our further analyses show that UniVL-DR can alleviate overfit single-modality signals by balancing hard negatives during training and bridging the modality gap between vision and language by verbalizing images. All experimental results show that learning one universal representation space is starting to benefit single-modality tasks—pretraining representation models on multi-modality and using our techniques can learn additional signals from multi-modalities, overcome the modality boundary, and provide convincing gains in single/multi-modality tasks.

## 2 RELATED WORK

Document retrieval is a typical single modality retrieval task, which aims to return related documents for user queries and can be tackled with dense retrievers (Xiong et al., 2021b; Lewis et al., 2020; Zhan et al., 2021; Li et al., 2021b; Yu et al., 2021). Dense retrievers encode queries and documents with pretrained language models (Devlin et al., 2019) and map them in an embedding space to conduct an efficient search. The query and document encoders are usually contrastively trained with in-batch negatives, BM25 retrieved negatives, and hard negatives (Karpukhin et al., 2020; Xiong et al., 2021a).

Recently, lots of work has focused on multi-modal retrieval tasks, which retrieve texts and images to satisfy the multi-modality information needs of users (Hannan et al., 2020; Singh et al., 2021; Talmor et al., 2021; Chang et al., 2022). WebQA (Chang et al., 2022), an open-domain multi-modal question answering benchmark, is built to encourage the following work to represent multi-modal knowledge in a unified space and answer user queries with the information from attribute modalities. It is a more realistic setting, which avoids synthesizing queries with templates (Talmor et al., 2021) and downplays the role of modality disambiguation (Hannan et al., 2020) in the multi-modality modeling.

To search information from large-scale multi-modality sources, WebQA (Chang et al., 2022) employs a divide-and-conquer pipeline to search text and image candidates with BM25 and CLIP (Radford et al., 2021) and then fuse these retrieval results using a vision-language model. However, single-

**Query:** how many years did william bradford serve as governor of plymouth colony?
**Retrieval Candidates:**

**Text1:** William Bradford was an English Separatist leader in Leiden and Holland. He served as Plymouth Colony Governor five times covering about thirty years between 1621 and 1657.
**Text2:** William Bradford was a passenger on the Mayflower in 1620. He travelled to the New World to live.
**Text3:** William Bradford was the governor of Plymouth Colony for 30 years. The colony was founded by people called Puritans.

**Query:** A woman wearing a net on her head cutting a cake.
**Retrieval Candidates:**

Image Retrieval ⇧
**Query:**                                    Text Retrieval ⇩

**Retrieval Candidates:**
**Text1:** A woman wearing a net on head cutting a cake.
**Text2:** A baker woman preparing bread dough on a tray with wax paper.

**Query:** What water-related object is sitting in front of the Torre del Reloj?
**Retrieval Candidates:**
Image1     Image2     Image3

**Text1:** The Torre del Reloj Spanish is the main city gate of the historic center of Cartagena de Indias.
**Text2:** The Torre del Reloj is the clock tower, known as Arquillo Clock, and is one of the most emblematic buildings of Chiclana.
**Text3:** Other landmarks in the city include the Torre del Reloj (Clock Tower).

(a) Text-Text Retrieval.  (b) Cross Modality Retrieval.  (c) Multi-Modal Retrieval.

Figure 2: Examples of Different Retrieval Tasks.

modality retrievers, such as BM25 and CLIP, usually show distinct retrieval effectiveness (Chang et al., 2022), leading to modality discrimination during fusing retrieval results from different modalities.

When building a unified multi-modal retriever, vision-language pretraining (VLP) is crucial to learn universal representations for texts and images, which has also shown success on lots of vision-language benchmarks (Uppal et al., 2022; Han et al., 2020; Khan et al., 2021; Du et al., 2022). Most VLP approaches encode texts and images and pretrain encoders with two tasks: masked token prediction and text-image matching (Zhang et al., 2021). These VLP methods teach vision-language models to learn the semantic alignments between texts and images, as well as encode images with the regional features of detected objects (Chen et al., 2019; Lu et al., 2019; Tan and Bansal, 2019; Su et al., 2020; Li et al., 2019; 2021a; Cho et al., 2021; Hu et al., 2020; Gan et al., 2020) or the whole image features (Xu et al., 2021; Kim et al., 2021; Huang et al., 2021b; Wang et al., 2021).

# 3    MULTI-MODAL RETRIEVAL TASK

As shown in Figure 3, we compare different retrieval tasks and tell apart the differences between multi-modal retrieval and other two tasks, single modality retrieval and cross modality retrieval.

**Single Modality Retrieval.** Single modality retrieval focuses on conducting relevance searching in one modality space, which includes text-text retrieval and image-image retrieval. Text-text retrieval (Bajaj et al., 2016) aims to search relevant candidates from the text collection $\mathcal{T} = \{T_1, ..., T_n\}$ to answer a query $q$. And image-image retrieval (Yoon et al., 2021) focuses more on returning similar images from the image collection $\mathcal{I} = \{I_1, ..., I_m\}$ for the given image $I_j$.

**Cross Modality Retrieval.** The cross modality retrieval, e.g. MSCOCO (Chen et al., 2015) and Flickr30K (Young et al., 2014), contains two subtasks: text-image retrieval and image-text retrieval. Given an image caption $T_i$ or an image $I_j$, these tasks require retrieval models to conduct cross-modality matching between images and captions, aiming to search candidates from images $\mathcal{I} = \{I_1, ..., I_m\}$ or image captions $\mathcal{T} = \{T_1, ..., T_n\}$, respectively. Such cross-modality interactions are built to align semantics between captions and images, which is distinct from the search relevance.

**Multi-Modal Retrieval.** Given a query $q$, the multi-modal retrieval task (Chang et al., 2022) helps users uncover the information from multi-modality sources $\mathcal{D} = \{T_1, ..., T_n, I_1, ..., I_m\}$.

Different from single/cross modality retrieval, multi-modal retrieval aims at returning relevant candidates from the multi-modality documents $\mathcal{D}$. The retrieval results may consist of texts, images, or a mixture of them according to user query $q$. Different from existing text and sketch base image retrieval (Sangkloy et al., 2022; Dutta and Akata, 2020; Dey et al., 2018; Mai et al., 2017), the multi-modal retrieval focuses more on relevance modeling between queries and documents, single/cross modality matching, and modality routing, making this task more challenging. Moreover, we can pre-route queries to a single modality and convert the multi-modal retrieval to two subtasks, text-text retrieval and text-image retrieval, which are single and cross modality retrieval tasks.

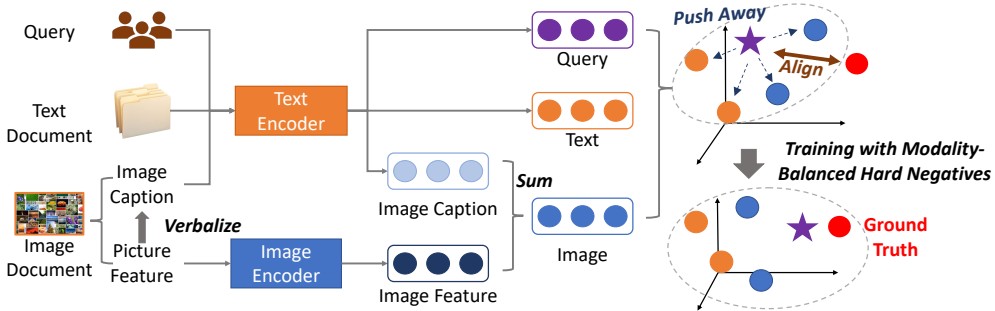

Figure 3: The Architecture of UniVL-DR.

# 4 UNIVSEARCH BY LEARNING A UNIFIED EMBEDDING SPACE

This section describes our Universal Vision-Language Dense Retrieval (UniVL-DR). As shown in Figure 3, given a query $q$ and multi-modality documents $\mathcal{D} = \{d_{\text{Text}}^1, ..., d_{\text{Text}}^n, d_{\text{Image}}^1, ..., d_{\text{Image}}^m\}$, it directly encodes query $q$, text document $d_{\text{Text}}^i$ and image document $d_{\text{Image}}^j$ in one embedding space, which conducts relevance modeling, modality routing, and result fusion in such a space (Sec. 4.1).

Texts and images usually have different understanding mechanisms, making it difficult to tackle multi-modality tasks. Nevertheless, language and vision can be commonly translated as a type of mentalese to better communicate between different modules in our brains (Cavanagh, 2021), thus a unified representation method has the ability to break the boundary of different modalities and benefit vision-language learning. To build a unified multi-modal retrieval system, UniVL-DR learns a universal embedding space by contrastively optimizing vision-language representations using hard negatives with balanced-modality sampling (Sec. 4.2) and bridging the modality gap via verbalizing the picture to paraphrase pixel semantics in the raw text space (Sec. 4.3).

## 4.1 MULTI-MODALITY DENSE RETRIEVAL

UniVL-DR gets representations of queries, image documents and text documents with two encoders: *TextEnocder* and *ImgEncoder*. Specifically, the image document $d_{\text{Image}}^j$ consists of a picture $I_j$ and an image caption $C_j$, thus we utilize *ImgEncoder* and *TextEnocder* to encode $I_j$ and $C_j$.

**Query Encoding.** UniVL-DR directly encodes the query $q$ to get its representation $\vec{q}$:
$$\vec{q} = TextEnocder(q). \tag{1}$$

**Text Document Encoding.** To represent text documents, UniVL-DR also leverages the *TextEnocder* to encode the $i$-th text document $d_{\text{Text}}^i$ as $\vec{d}_{\text{Text}}^i$:
$$\vec{d}_{\text{Text}}^i = TextEnocder(d_{\text{Text}}^i). \tag{2}$$

**Image Document Encoding.** Different from text documents, image documents can be represented by picture features and image captions and the textual captions can help better understand the semantics of image documents (Baldrati et al., 2022). Thus, UniVL-DR encodes picture $I_j$ and image caption $C_j$ and then sums these embeddings to get the representation $\vec{d}_{\text{Image}}^j$ of $j$-th image document:
$$\vec{d}_{\text{Image}}^j = ImgEnocder(I_j) + TextEnocder(C_j). \tag{3}$$

The representations $\vec{d}_{\text{Image}}^j$ and $\vec{d}_{\text{Text}}^i$ of image document and text document use the same $TextEnocder$ to encode their textual information, which bridges different modalities in the text space and helps to build a universal embedding space for multi-modality retrieval.

**Multi-modality Document Retrieval.** The cosine similarity score $f(q, d)$ of query $q$ and document candidate $d \in \mathcal{D}$ can be calculated to estimate the relevance between $q$ and $d$:
$$f(q, d) = \cos(\vec{q}, \vec{d}), \tag{4}$$

where $\vec{q}$ and $\vec{d}$ are the representations of $q$ and $d$. The efficient similarity calculation between queries and the multi-modality documents can be provided by FAISS (Johnson et al., 2019).

## 4.2 Universal Representation Learning

UniVL-DR employs a vision-language model, CLIP (Radford et al., 2021), to learn universal representations for queries and multi-modality documents, which is knowledgeable about cross-modality retrieval. UniVL-DR optimizes the universal embedding space through training with modality-balanced hard negatives, which avoids overfitting to the signals of single-modality during multi-modal co-training.

Given the query $q$ and its relevant candidate $d^+ \in \mathcal{D}$, the embedding space can be optimized by sampling hard negatives $\mathcal{D}^-$ and minimizing the following contrastive training loss $L$:

$$
\begin{aligned}
L &= -\log \frac{e^{f(q,d^+)/\tau}}{e^{f(q,d^+)/\tau} + \sum_{d^- \in \mathcal{D}^-} e^{f(q,d^-)/\tau}} \\
&= -\underbrace{f(q,d^+)/\tau}_{L_{\text{Align}}} + \log(e^{f(q,d^+)/\tau} + \underbrace{\sum_{i=1}^{k_1} e^{f(q,d^{i-}_{\text{Image}})/\tau}}_{L_{\text{Image}}} + \underbrace{\sum_{j=1}^{k_2} e^{f(q,d^{j-}_{\text{Text}})/\tau}}_{L_{\text{Text}}}),
\end{aligned}
\tag{5}
$$

where $\tau$ is the temperature to scale the similarity score. During training, we in fact maximize $L_{\text{Align}}$ and minimize $L_{\text{Image}}$ and $L_{\text{Text}}$, which make queries closer to related documents and away from unrelated documents. If $k_1 > k_2$ or $k_2 > k_1$, we can achieve a smaller loss $L_{\text{Image}} + L_{\text{Text}}$ by simply making queries far away from the image collection or the text collection. Such a behavior can win a lower loss $L$ but overfits the ranking features from single/cross modality matching, leading to a modality discrimination during retrieval. Our modality-balanced negative training strategy keeps $k_1 = k_2 = k$ to better train the modality selection ability of retrievers.

## 4.3 Image Verbalization for Expansion

UniVL-DR provides another way to bridge the modality gap between texts and images by verbalizing picture pixel features, including image caption and query generation methods.

Following Li et al. (2020), we can represent a picture $I_j$ using detected objects $\mathcal{O} = \{O_1, ..., O_l\}$. For each image object $O_i$, we can get its pixel feature $\vec{O}_i$ and the predicted class $\hat{O}_i$. Then UniVL-DR uses a vision-language model, such as VinVL (Zhang et al., 2021), to verbalize image documents. Specifically, we generate potentially matched captions or related queries as the image verbalization results $V(I_j)$, according to the picture $I_j$ or the image document $d^j_{\text{Image}} = \{I_j, C_j\}$.

We can first feed the predicted classes $\{\hat{O}_1; ...; \hat{O}_l\}$ and regional features $\{\vec{O}_1; ...; \vec{O}_l\}$ of detected objects into image verbalization models. Then we train the model to generate image caption $C_j$:

$$
X_j^c = [\text{CLS}]; C_j; [\text{SEP}]; \hat{O}_1; ...; \hat{O}_l; [\text{SEP}]; \vec{O}_1; ...; \vec{O}_l;
\tag{6}
$$

or replace the detected object classes $\{\vec{O}_1; ...; \vec{O}_l\}$ in the input sequence $X_j^c$ with the image caption $C_j$ to generate related query $q$ of the image document $d^j_{\text{Image}}$:

$$
X_j^q = [\text{CLS}]; q; [\text{SEP}]; C_j; [\text{SEP}]; \vec{O}_1; ...; \vec{O}_l,
\tag{7}
$$

where ; is the concatenation operation, and [CLS] and [SEP] are special tokens. During training or inference, we utilize Masked Language Modeling (MLM) (Devlin et al., 2019) to mask and predict some or all of the tokens of image caption $C_j$ and query $q$ in the inputs $X_j^c$ and $X_j^q$, aiming to train image verbalization models or generate verbalized captions and queries.

Finally, we enhanced the representations of image documents by expending their text representations $C_j^*$ by expanding the raw caption $C_j$ with image verbalization results $V(I_j)$:

$$
C_j^* = C_j; [\text{SEP}]; V(I_j),
\tag{8}
$$

where the enhanced text representation $C_j^*$ is used to replace the raw caption $C_j$ in E.q. 3 during encoding the image document $d^j_{\text{Image}}$.

## 5  EXPERIMENTAL METHODOLOGY

This section describes the dataset, baselines, some vision language models used in our experiments, and implementation details.

**Dataset.** A multi-hop and multi-modal open domain question answering dataset WebQA (Chang et al., 2022) is used in our experiments. We process the WebQA dataset in an open domain retrieval setting and show the details in Appendix A.1.

**Evaluation Metrics.** We use NDCG@$K$, MRR@$K$, Recall@20, and Recall@100 as the evaluation metrics. $K$ can be 10 and 20. And we regard MRR@10 as our main evaluation (Bajaj et al., 2016).

**Vision-Language Models.** In our experiments, we employ two state-of-the-art vision-language models, VinVL (Zhang et al., 2021) and CLIP (Radford et al., 2021) to implement different retrieval models in our experiments. VinVL (Zhang et al., 2021) inherits Oscar (Li et al., 2020) architecture, which extracts object tags and region features to represent images, and learns cross-modal representations by aligning semantics between images and texts. Different from VinVL, CLIP (Radford et al., 2021) utilizes a dual encoder to project images and texts in the same semantic space for computing their similarity scores and is trained on a large-scale dataset WebImageText that contains 400 million image-text pairs. It has shown strong effectiveness in cross-modality retrieval.

**Baselines.** Our baselines contain several models in the settings of single modality retrieval, divide-and-conquer, and universal multi-modal retrieval.

*Single modality retrieval.* In this setting, we represent image documents with captions and employ text retrievers, BM25 and DPR (Karpukhin et al., 2020) as baselines. DPR is trained with NQ (Kwiatkowski et al., 2019), which is similar to the textual source of WebQA. Then we continuously train DPR with in-batch and hard negatives to implement NQ-DPR and NQ-ANCE models.

*Divide-and-conquer.* We first employ three widely used retrievers, BM25, VinVL-DPR, and CLIP-DPR, to conduct text-text retrieval and text-image retrieval. Then the multi-modality retrieval results are fused according to their uni-modal rank reciprocals or oracle modality routing. The latter one shows the upper bound of the retrieval performance of our divide-and-conquer models.

*Multi-modal retrieval.* In our experiments, we also build two multi-modal retrieval baselines: VinVL-DPR and CLIP-DPR. VinVL-DPR and CLIP-DPR represent image documents with caption and picture features. And then they optimize VLP models, VinVL (Zhang et al., 2021) and CLIP (Radford et al., 2021), with in-batch negatives to learn universal representations for multi-modal retrieval.

**Implementation Details.** During training UniVL-DR, we employ the text and image encoders from CLIP, truncate the text with the max length of 77[1] and set the batch size to 64, learning rate=$5e-6$, max training epoch to 20, and the temperature hyperparameter $\tau = 0.01$. In our experiments, we retrieve Top 100 documents using CLIP-DPR and sample two hard negatives of different modalities ($k = 1$) from these candidates. All models are tuned with AdamW optimizer, are evaluated per 500 steps, and set early stop step as 5. More experimental details are shown in Appendix A.2.

## 6  EVALUATION RESULTS

In this section, we study the performance of UniVL-DR, its advantages in multi-modal retrieval, the effectiveness of our modality-balanced hard negative training strategies, and how our image verbalization methods bridge the modality gap between texts and images.

### 6.1  OVERALL PERFORMANCE

The multi-modal retrieval performance of different models is shown in Table 1.

Our UniVL-DR outperforms all baselines with more than 7% improvement on ranking evaluation, recalls more than 6% relevant multi-modality documents, and even outperforms the divide-and-conquer model guided by oracle modality routing. Such significant improvements illustrate the effectiveness of UniVL-DR in building a multi-modal retrieval system.

---

[1] https://github.com/openai/CLIP

| Setting | Model | MRR@10 | NDCG@10 | MRR@20 | NDCG@20 | Rec@20 | Rec@100 |
|---|---|---|---|---|---|---|---|
| Single Modality (Text Only) | BM25 | 53.75 | 49.60 | 54.10 | 51.72 | 68.16 | 80.69 |
| | DPR (Zero-Shot) (Karpukhin et al., 2020) | 22.72 | 20.06 | 23.14 | 21.79 | 32.78 | 45.43 |
| | CLIP (Zero-Shot) (Radford et al., 2021) | 18.16 | 16.76 | 18.60 | 18.27 | 27.97 | 39.83 |
| | BERT-DPR (Karpukhin et al., 2020) | 42.16 | 39.57 | 42.76 | 42.26 | 60.85 | 77.10 |
| | NQ-DPR (Karpukhin et al., 2020) | 41.88 | 39.65 | 42.44 | 42.35 | 61.71 | 78.57 |
| | NQ-ANCE (Xiong et al., 2021a) | 45.54 | 42.05 | 45.93 | 43.83 | 58.42 | 69.31 |
| Divide-Conquer | VinVL-DPR | 22.11 | 22.92 | 22.80 | 25.41 | 46.27 | 62.82 |
| | CLIP-DPR | 37.35 | 37.56 | 37.93 | 40.77 | 69.38 | 85.53 |
| | BM25 & CLIP-DPR | 42.27 | 41.58 | 42.79 | 44.69 | 73.34 | 87.50 |
| | BM25 & CLIP-DPR (Oracle Modality) | 61.05 | 58.18 | 61.37 | 60.45 | **80.82** | **90.83** |
| UnivSearch | CLIP (Zero-Shot) | 10.59 | 8.69 | 10.80 | 9.52 | 14.32 | 20.21 |
| | VinVL-DPR | 38.14 | 35.43 | 38.74 | 37.79 | 53.89 | 69.42 |
| | CLIP-DPR | 48.83 | 46.32 | 49.34 | 49.11 | 69.84 | 86.43 |
| | UniVL-DR | **62.40** | **59.32** | **62.69** | **61.22** | 80.37 | 89.42 |

Table 1: Multi-Modal Retrieval Performance. VinVL-DPR, CLIP-DPR, NQ-DPR and BERT-DPR are trained with in-batch negatives, while NQ-ANCE is trained with hard negatives.

Similar to UniVL-DR, BM25 learns universal textual representations for image/text documents and shows strong ranking effectiveness. To build a divide-and-conquer system, we use BM25 and CLIP-DPR to implement text-text and text-image retrievers and then fuse the results from different retrievers. With the help of oracle modality routing, the divide-and-conquer system shows better ranking results and recalls more relevant documents than BM25. Nevertheless, this system shows a distinct performance when using the uni-modal rank reciprocals to route queries, showing the challenge of fusing retrieval results in divide-and-conquer. CLIP-DPR and UniVL-DR can deal with this problem by learning universal representations for queries and multi-modality documents, which unifies the multi-modality relevance modeling and retrieval result fusion. Thanks to our multi-modality training strategies, UniVL-DR achieves more than 10% improvement on multi-modal retrieval than CLIP-DPR. The following experiments further explore how UniVL-DR learns universal representations for multi-modal retrieval and bridges the gap between images and texts.

## 6.2 ABLATION STUDIES

The ablation studies are conducted to study model performance on multi-modal retrieval. And we also evaluate the effectiveness of UniVL-DR on both text-text and text-image retrieval tasks, which aims at showing the influence of multi-modal learning on these single/cross modality retrieval tasks.

As shown in Table 2, we evaluate the retrieval effectiveness of different vision-language models, VinVL-DPR and CLIP-DPR. They are trained with in-batch negatives on text-text/image and multi-modal retrieval tasks. In the single/cross modality setting, we fine-tune vision-language models with a group of queries that only contain related documents in text modality or image modality. Our multi-modality training setting uses all queries to train these vision-language models and equally samples in-batch negatives from the documents of different modalities.

For both CLIP-DPR and VinVL-DPR, image captions are usually more effective to represent image documents than figure features, which demonstrates the difficulty in understanding figure semantics with only figure pixels. Thus, UniVL-DR tries to verbalize the figure features by extracting the objects that appear in the figure and describing the figure facts among detected objects (Zhang et al., 2021). The image verbalization results paraphrase picture pixel facts in natural language and help to enhance the textual representations of images by expanding image verbalization results to image captions. As a result, UniVL-DR uses such an enhanced text representation for image documents and then employs the same module to encode text information of image documents and text documents. It helps to build universal representations for multi-modality documents by breaking the modality boundary and fully using additional training signals from different modalities, making UniVL-DR achieve the best retrieval performance on multi-modal retrieval among all baseline models.

UniVL-DR also shows its advantages by outperforming all baseline models on both text-text and text-image retrieval tasks, demonstrating that multi-modality modeling indeed benefits single/cross modality retrieval. In the multi-modal retrieval setting, CLIP-DPR is converted from a text-text retriever to a multi-modal retriever after adding figure features. CLIP-DPR achieves better performance on the text-image retrieval task than CLIP-DPR w/o figure feature, which illustrates that image features provide additional signals to help multi-modality models distinguish related image documents. On the contrary, the multi-modal retrieval performance of CLIP-DPR is decreased,

| Model | Retrieval Performance | | |
|---|---|---|---|
| | Text | Image | Multi |
| *Single/Cross Modality Retrievers* | | | |
| BERT-DPR | 37.09 | 52.34 | - |
| VinVL-DPR w/o caption | - | 3.67 | - |
| VinVL-DPR w/o fig feature | - | 51.56 | - |
| VinVL-DPR | 25.00 | 48.68 | - |
| CLIP-DPR w/o caption | - | 17.74 | - |
| CLIP-DPR w/o fig feature | - | 58.17 | - |
| CLIP-DPR | 52.57 | 59.95 | - |
| *Universal Multi-Modal Retrievers* | | | |
| VinVL-DPR w/o fig feature | 29.01 | 46.55 | 36.13 |
| VinVL-DPR | 29.95 | 49.65 | 38.14 |
| CLIP-DPR w/o fig feature | 51.47 | 57.36 | 50.33 |
| CLIP-DPR | 51.75 | 60.61 | 48.83 |
| UniVL-DR | **60.72** | **65.57** | **62.40** |

Table 2: Retrieval Performance of Different Ablation Models. MRR@10 is used as the evaluation metric.

| Sampling | Retrieval Performance | | | Retrieved |
|---|---|---|---|---|
| | Text | Image | Multi | Image (%) |
| *In-batch Training* | | | | |
| CLIP-DPR (Random) | 51.75 | 60.61 | 48.83 | 26.82 |
| Balanced In-batch | 52.24 | 59.99 | 49.88 | 30.35 |
| *Hard Negative Training* | | | | |
| Only Texts | 54.92 | 52.88 | 36.26 | 91.74 |
| Only Images | 55.85 | **66.51** | 33.49 | 1.97 |
| 2 Texts & 1 Image | 59.18 | 65.15 | 61.64 | 49.53 |
| 1 Text & 2 Images | 57.86 | 66.23 | 61.20 | 47.88 |
| ANCE (Random) | 59.85 | 64.80 | 61.72 | 50.01 |
| Balanced In-batch | **60.58** | 65.21 | **62.29** | 49.11 |

Table 3: Effectiveness of Different Hard Negative Training Strategies. These hard negative trained models start from in-batch trained ones and are continuously fine-tuned with different numbers of hard negatives. The MRR@10 score and the image ratio of the Top 10 retrieved candidates are shown to evaluate the modality preference.

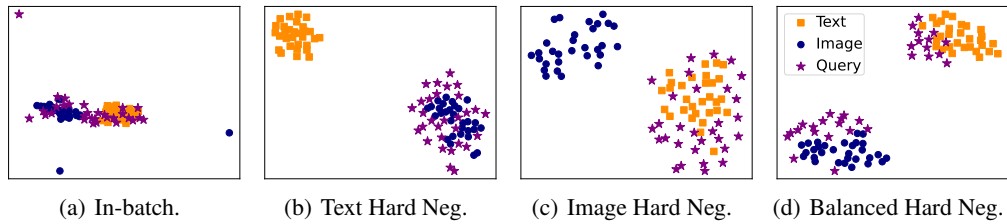

(a) In-batch.  (b) Text Hard Neg.  (c) Image Hard Neg.  (d) Balanced Hard Neg.

Figure 4: Embedding Space Visualization of Different Training Strategies. We randomly sample 30 image documents, 30 text documents and 30 queries. And then we visualize their embeddings learned by CLIP-DPR and the continuously trained models with different hard negative sampling strategies.

showing that CLIP-DPR fails to fuse retrieval results from different modalities. UniVL-DR uses a modality-balanced hard negative training strategy to learn universal representations for queries and documents, which deals with the challenge of fusing retrieval results, helps to achieve more gains on the multi-modal retrieval task, and enhances the modality disambiguation ability.

### 6.3 Effectiveness of Balanced Hard Negative Sampling

In this experiment, we study the training strategies of UniVL-DR that are used in learning universal multi-modality representations and show the effectiveness of different negative sampling methods.

As shown in Table 3, we start from the multi-modal retriever CLIP-DPR, continuously fine-tune it with different hard negative sampling methods, and show their performance on different retrieval tasks. Our experimental results show that the in-batch trained models prefer to return text documents than image documents as the ranking results, even the image-answerable queries take a larger portion (about 51.6%) in the training data. It illustrates that training multi-modality retrievers with modality-unbalanced negatives usually leads to undesired modality bias during retrieval.

Then we continuously train CLIP-DPR with hard negatives sampled from top-retrieved multi-modality results from CLIP-DPR and significantly improve its retrieval performance in all testing scenarios. Our modality-balanced hard negative sampling strategy achieves the best retrieval performance among all negative sampling methods, showing its important role in building a universal multi-modal retrieval model. Compared with ANCE (Random), our modality-balanced sampling strategy mitigates the modality variance during contrastive training and provides more useful signals to train the modality disambiguation ability of universal multi-modal retrievers.

Finally, we visualize the embedding space of different retrieval models in Figure 4. After training with modality-balanced hard negatives, UniVL-DR learns a more uniform and effective embedding space for multi-modal retrieval. In this embedding space, both text and image documents are

| Model | In-batch Training | | | Hard Neg Training | | |
|---|---|---|---|---|---|---|
| | Text | Image | Muti | Text | Image | Muti |
| UniVL-DR | 51.75 | **60.61** | 48.83 | 60.58 | 65.21 | 62.29 |
| w. Verbalized Caption | 51.51 | 60.57 | 49.49 | 59.86 | **65.87** | 62.24 |
| w. Verbalized Query | **52.14** | 59.72 | **50.21** | **60.72** | 65.57 | **62.40** |

Table 4: Performance of Multi-Modality Retrieval Models with Different Image Verbalization Methods. All models are evaluated with MRR@10.

assigned in different areas of the embedding space, and queries are routed to different areas for returning documents from corresponding modalities. As shown in Figure 4(b) and Figure 4(c), when the retrieval models are only trained with hard negatives of text and image documents, the query embeddings are concentrated and respectively assigned closer to the areas of image and text documents. It demonstrates that multi-modality retrieval model usually overfits the training signals of in-batch majority modality to win a lower contrastive loss during training. UniVL-DR alleviates this problem by balancing the modalities of hard negatives in contrastive training.

### 6.4 Bridging Cross-Modality Matching with Image Verbalization

UniVL-DR uses image verbalization methods to generate matched captions or related queries to bridge the modality gap between texts and images. In this experiment, we show the effectiveness of different image verbalization strategies on text-text, text-image, and multi-modal retrieval tasks.

As shown in Table 4, our image verbalization methods demonstrate their ability on enhancing the text representations of image documents by achieving better text-image retrieval results. These image verbalization methods aim to generate informative text clues to help retrievers distinguish the query-related image documents in the text space. Then the text-text and multi-modal retrieval performance is also improved with the help of verbalized captions or queries, showing the effectiveness of our image verbalization methods in bridging the modality gap between images and texts and benefiting the single modality tasks using additional training signals from different modalities.

Compared with verbalized captions, our query verbalization method aligns the necessary semantics in captions, e.g. mentioned entities, with image objects and verbalizes the figure pixels with the help of caption semantics. Enhancing image representations using verbalized queries usually achieves better retrieval effectiveness than using verbalized captions. It showcases that our query verbalization method can provide more meaningful text clues for relevance modeling and multi-modality learning. Moreover, some additional experiments are provided to study the effectiveness of different image verbalization methods. We first show the relationship between the effectiveness of verbalized queries and manual caption lengths in Appendix A.4 and then conduct some case studies in Appendix A.5 to explore the characteristics of different image verbalization methods.

## 7 Conclusion

This paper proposes UniVL-DR, which models singe/cross modality matching and retrieval result fusion in one universal embedding space. UniVL-DR proposes an effective multi-modality training strategy to learn universal representations for queries and documents, which breaks the modality boundary between vision and language, and helps to achieve state-of-the-art multi-modal retrieval performance. Our experiments show that UniVL-DR can bridge the modality gap with image verbalization technologies and avoid overfitting the training signals of one modality by optimizing retrievers with modality-balanced hard negatives.

## Acknowledgments

This work is supported by Beijing Academy of Artificial Intelligence (BAAI), the Natural Science Foundation of China under Grant No. 62206042, No. U1811261 and No. 62006129, the Fundamental Research Funds for the Central Universities under Grant No. N2216013, China Postdoctoral Science Foundation under Grant No. 2022M710022, and National Science and Technology Major Project (J2019-IV-0002-0069).

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

| Modality | #Documents | #Queries | | | Task Category |
|---|---|---|---|---|---|
| | | Train | Dev | Test | |
| Image | 389,750 | 16,400 | 2,554 | 2,511 | Text-Image Retrieval |
| Text | 787,697 | 15,366 | 2,446 | 2,455 | Text-Text Retrieval |
| Total | 1,177,447 | 31,766 | 5,000 | 4,966 | Multi-Modal Retrieval |

Table 5: Data Statistics of Vision-Language QA Benchmark WebQA in Open-Domain Setting.

# A APPENDIX

## A.1 DATA STATISTICS

A multi-hop and multi-modal open domain question answering dataset WebQA (Chang et al., 2022) is used in our experiments. The dataset contains images and passages that are crawled from the general Web and Wikipedia. In our experiments, we randomly sample 5,000 queries from the original training set of WebQA as the development set for evaluation. All data statistics are shown in Table 5. To build an open-domain benchmark, we collect 389,750 images and 787,697 texts as multi-modal retrieval sources. The image collection contains all images collected by the WebQA dataset, while the text collection contains all relevant passages of all 41,732 queries, which are Wikipedia snippets selected by matching noun chunks in the queries (Chang et al., 2022).

## A.2 ADDITIONAL EXPERIMENT DETAILS

This subsection describes additional implementation details. In our experiments, we employ two pretrained vision-language models, VinVL (Zhang et al., 2021) and CLIP (Radford et al., 2021), and the pretrained language model, BERT (Devlin et al., 2019) to implement different retrieval models.

*VinVL-DPR.* For VinVL variant models, we first detect the image objects and extract corresponding region features following VinVL[2]. Then we concatenate image captions and image region features as inputs to feed into VinVL models and get the image representations. We initialize VinVL with the checkpoint trained on the MSCOCO image retrieval task and continuously train the model on the WebQA dataset with in-batch negatives. During training, we set the batch size to 32, learning rate=$2e-5$, accumulate step as 1, and max training epoch to 30. We truncate the queries, image captions, text documents, and image region features with max lengths of 70, 70, 200, and 50.

*CLIP-DPR.* For training CLIP-DPR, we start from the ViT-B/32 version of CLIP and continuously train CLIP on the WebQA dataset with in-batch negatives. We truncate texts with the max length of 77 and set accumulate step as 1, batch size to 64, learning rate=$5e-6$, max training epoch to 20, and the temperature hyperparameter $\tau = 0.01$. The cosine annealing strategy is used to schedule the learning rate during training.

*BERT-DPR.* We initialize our retriever with the bert-base-uncased checkpoint, which is provided by Hugginface Transformers[3]. During training, we set the batch size to 32, learning rate=$5e-5$, accumulate step as 1, and max training epoch to 30. We truncate the queries, text documents, and image captions with max lengths of 70, 200, and 70.

*NQ-DPR/NQ-ANCE.* NQ-DPR and NQ-ANCE start from the NQ-trained DPR model (Karpukhin et al., 2020), which uses a dual encoder architecture to encode queries and documents. All experimental settings keep the same with BERT-DPR. Besides, NQ-ANCE is tuned with the hard negatives sampled from the Top 100 retrieved candidates of NQ-DPR Xiong et al. (2021a).

## A.3 EXPERIMENTAL DETAILS OF IMAGE VERBALIZATION

The image verbalization models are used to generate potentially matched captions or related questions for an image. Our experiments start from the image caption generation model, which is trained on the MSCOCO image caption task (Zhang et al., 2021) to generate related captions or queries to verbalize images.

---

[2]https://github.com/microsoft/scene_graph_benchmark
[3]https://github.com/huggingface/transformers

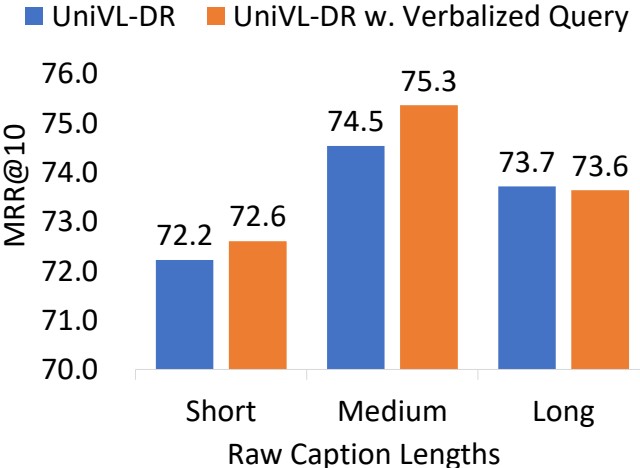

Figure 5: Performance of Multi-Modal Retrieval Models with Different Caption Lengths. The query lengths of short, medium and long groups are in the sections of [0, 10), [10, 20), and [20, $+\infty$).

We can first directly generate image-related captions as the image verbalization results using the image caption model provided by VinVL (Zhang et al., 2021). As shown in E.q. 6, we first detect the image objects in the images and then feed the predicted classes and region features of detected objects to the VinVL model. In our experiments, we fix the parameters of the VinVL-based image caption model and generate the caption for each image.

During generating image-related queries, as shown in E.q. 7, we concatenate the image-related query, image caption, and image regional features as the input. We continuously train the VinVL-based image caption model by randomly masking the tokens in queries, and optimizing vision-language models to fill in the masked positions. Different from image caption models, our query generation method tries to align the semantics in image captions and image pixel features instead of mapping the predicted classes and regional image features of detected objects, which can help vision-language models better understand the image semantics (Huang et al., 2021a).

During training and inference, we set the generated tokens up to 20 and the beam size to 5. We truncate the queries, image captions, and image region features with the max lengths of 40, 30, and 50, respectively. The mask probability is set to 0.15. More experimental details can be referred to Zhang et al. (2021).

## A.4 IMAGE VERBALIZATION PERFORMANCE WITH DIFFERENT CAPTION LENGTHS

In this subsection, we evaluate the multi-modal retrieval performance of UniVL-DR with different verbalized queries. Specifically, we evaluate the effectiveness of image-verbalized queries in the multi-modal retrieval task. These image-verbalized queries are generated with the image documents with different manual caption lengths.

We group the testing examples into three categories according to the manual caption lengths of the image documents and calculate the average MRR@10 score for each group. The ratios are 42.33%, 36.84%, and 20.83% of the short, medium, and long caption length groups.

As shown in Figure 5, the experimental results show that our query generation method mainly helps to improve the retrieval effectiveness on the queries of short length and medium length, illustrating that these generated queries can provide some crucial textual clues in image representations of shorter captions. These expanded text clues help retrieval models better understand image semantics, more effectively represent images via enhanced textual information, and conduct cross-modality matching more easily. Moreover, the queries in the medium caption length group achieve the best performance, because the image captions of medium lengths can cover more necessary text clues for generating more informative verbalization results.

| Figures | Text |
|---|---|
| | **Query:** Does a Minnetonka Rhododendron flower have petals in a cup shape? 

 Manual Caption: 2020-05-08 15 17 05 Minnetonka Rhododendron flower along Tranquility Court in the Franklin Farm section of Oak Hill, Fairfax County, Virginia Minnetonka Rhododendron flower along Tranquility Court in the Franklin Farm section of Oak Hill, Fairfax County, Virginia 
 Verbalized Caption: a purple flower with green leaves and purple flowers. 
 Verbalized Query: what shape are the petals of the minnetonka rhododendron flower? |
| | **Query:** Are the heads of Iranian women covered in traditional clothing? 

 Manual Caption: Iranian family, gathered together wearing traditional clothes - Nishapur - Nowruz2014 Iranian family, gathered together wearing traditional clothes 
 Verbalized Caption: a group of people in costumes standing in a park. 
 Verbalized Query: how many people are wearing hats in the group of iranian family members? |
| | **Query:** At the 1928 Amsterdam Olympics, what is the maximum number of buttons that you can get on the Egyptian men's uniform? 

 Manual Caption: Egyptische atleten bij OS Amsterdam 1928 - Egyptian Olympic athletes, Amsterdam 1928 (6941436605) http://www.spaarnestadphoto.nl/component/option,com memorix ... 
 Verbalized Caption: a group of men in suits and hats standing in a field. 
 Verbalized Query: did all the men in the egyptian olympic athletes wear the same type of caps? |
| | **Query:** What water-related object is sitting in front of the Torre del Reloj? 

 Manual Caption: Torre del Reloj de la Plaza Colón de Antofagasta (1) 
 Verbalized Caption: a water fountain in front of a clock tower. 
 Verbalized Query: is the fountain in front of the clock tower at la plaza de reloj taller than |
| | **Query:** What color is the facade of bakery Sattin et Fils in Rethel, France? 

 Manual Caption: Rethel-FR-08-boulangerie Sattin-01 
 Verbalized Caption: a red storefront on a city street corner. 
 Verbalized Query: how many potted plants are outside of the boulangerie sattin? |
| | **Query:** Does the Durham Cathedral in England have any trees outside of it? 

 Manual Caption: Durham Cathedral, July 2014 (04) Durham Cathedral, Durham, County Durham, England 
 Verbalized Caption: a large building with two towers and a tree in front of it. 
 Verbalized Query: are there any trees near durham cathedral which are taller than the cathedral? |

Table 6: Image Verbalization Cases. We highlight the matching phrases between user queries and manual captions or image verbalization results.

## A.5 Case Studies on Different Image Verbalization Methods

This experiment shows some image verbalization cases in Table 6. We randomly sample queries that can be answered by image documents and show the manual captions, verbalized captions, and verbalized queries of the image documents.

Overall, these cases can be categorized into two groups according to the lengths of manual captions. The first three cases are longer and more informative to describe the image facts among mentioned objects, which can be directly used in text-image relevance modeling. On the contrary, the manual captions in the last three cases are written by the most representative entities that appeared in the images, making it difficult to distinguish the related images only according to these manual captions.

UniVL-DR employs two image verbalization methods to enhance the textual semantics of images. Generating image captions is the most intuitive way to paraphrase images with some pre-defined classes of image objects. Nevertheless, these object classes are too general and may be uninformative to provide semantics for matching, because the specific entities are critical to retrieving related documents in a question-answering system (Sciavolino et al., 2021). Different from these verbalized captions, the verbalized queries are usually more informative and meaningful and specify the image objects by copying entity names from the manual captions, such as the names of persons, places, and buildings. These entities can be directly matched with the given queries, which benefits cross-modality matching and helps to mitigate the modality gap between images and texts.

## A.6 Multi-Modal Retrieval with Different Image Representation Combination Methods

In this subsection, we conduct experiments to show the effectiveness of different methods in combining the representations of image captions and image features.

| Model | MRR@10 | MRR@20 | NDCG@10 | NDCG@20 |
|---|---|---|---|---|
| CLIP-DPR (Concatenation) | 23.18 | 23.61 | 20.74 | 22.94 |
| CLIP-DPR (Outer Product) | 39.89 | 40.47 | 37.40 | 40.22 |
| CLIP-DPR (Sum) | **48.83** | **46.32** | **49.34** | **49.11** |

Table 7: Multi-Modal Retrieval Performance with Different Representation Combining Methods. We concatenate, dot product and sum the representations of the image captions and image features to build different models.

| Model | MRR@1 | NDCG@5 | NDCG@10 | NDCG@20 |
|---|---|---|---|---|
| BM25 | 41.14 | 46.24 | 49.60 | 51.72 |
| BM25 & CLIP-DPR | 7.43 | 19.47 | 41.58 | 44.69 |
| BM25 & CLIP-DPR (Oracle Modality) | 46.19 | 54.20 | 58.18 | 60.45 |
| NQ-ANCE | 34.21 | 39.16 | 42.05 | 43.83 |
| CLIP-DPR | 34.92 | 42.30 | 46.32 | 49.11 |
| UniVL-DR | **47.56** | **55.28** | **59.32** | **61.22** |

Table 8: Additional Evaluations on Multi-Modal Retrieval Models.

As shown in Table 7, we evaluate the effectiveness of different combination models using CLIP-DPR. We concatenate, dot product (outer product), and sum the representations of image captions and image features to conduct three models: CLIP-DPR (Concatenation), CLIP-DPR (Outer Product), and CLIP-DPR (Sum).

CLIP-DPR (Sum) shows its effectiveness by achieving the best performance among all baselines. The sum operation is a commonly used semantic combination method, which is also used in BERT (Devlin et al., 2019) to combine token embedding and position embedding. On the contrary, the concatenation operation usually regards the representations of image captions and image features as subvectors and separates them into subspaces, making it hard to learn the semantics of image documents. On the other hand, the outer product operation conducts orthogonal representations during combining representations, which is not a typical combination method.

## A.7 ADDITIONAL EVALUATIONS ON MULTI-MODEL RETRIEVAL

In our experiments, we follow previous widely used retrieval benchmarks, MS MARCO (Bajaj et al., 2016) and BEIR (Thakur et al., 2021), and use NDCG@10/20 and MRR@10/20 to show the retrieval effectiveness of different retrieval models. MRR scores and NDCG scores are calculated by the MS MARCO official scripts[4] and TREC's evaluation tool[5].

As shown in Table 8, we also conduct some evaluations to show the retrieval performance of higher-ranked candidates using MRR@1 and NDCG@5. UniVL-DR also shows strong effectiveness by outperforming BM25 and CLIP-DPR with more than 6% improvements. Notably, UniVL-DR even shows better retrieval effectiveness than the BM25 & CLIP-DPR (Oracle Modality) model, which is in an ideal setting. It supports our claim that multi-modality modeling can also benefit single/cross-modality tasks.

## A.8 EXAMPLES OF HARD NEGATIVES

In this subsection, we randomly sample two queries and show some hard negatives in Figure 6, which are top-ranked documents using our CLIP-DPR model.

In the first case, when we ask "Is there greenery at Centennial Olympic Park?", the CLIP-DPR model can provide some image and text documents, which are regarded as hard negatives to continuously train dense retrievers. The negative images are about buildings, lawns, and trees, but these objects are not located at Centennial Olympic Park. Evidently, these negative images are on-topic with "greenery" but are not related to the given query. Training dense retrievers with these hard negatives can better teach retrievers to distinguish the subtle difference among these confusing images.

---

[4] https://github.com/microsoft/MSMARCO-Passage-Ranking/blob/master/ms_marco_eval.py
[5] https://github.com/cvangysel/pytrec_eval

**Query1:** Is there greenery at Centennial Olympic Park?

**Golden Doc**        **Image Doc1**        **Image Doc2**        **Image Doc3**

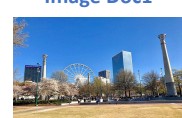 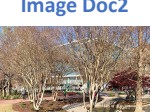 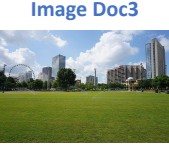

**Text Doc1:** The city's permanent memorial to the 1996 Olympics is Centennial Olympic Park, which was built as a focal point for the Games.

**Text Doc2:** Built as a legacy of the 1996 Olympic Games, Centennial Olympic Park, located on 21-acre (85,000 m2) area of Downtown, is the largest downtown park in the United States developed in the last 25 years.

**Text Doc3:** Centennial Olympic Park was designed as the "town square" of the Olympics, and thousands of spectators had gathered for a late concert and merrymaking.

**Query2:** The genus name Columba is Latin word means dove or what, which are particularly fond of roof spaces?

**Image Doc1**        **Image Doc2**        **Image Doc3**        **Image Doc4**

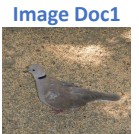 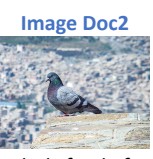 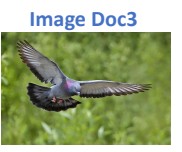 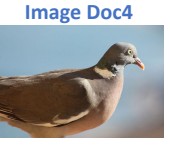

**Golden Doc:** *Pigeons are particularly fond of roof spaces. These often contain water tanks. Any water tank or cistern on a roof must, therefore, be secured and sealed off to keep the pigeons out of them. The popularity of a nesting area does not seem to be affected by the pigeons' population density.*

**Text Doc1:** The common name "columbine" comes from the Latin for "dove", due to the resemblance of the inverted flower to five doves clustered together.

**Text Doc2:** The genus name is Late Latin; falco derives from falx, falcis, a sickle, referring to the claws of the bird. The species name columbarius is Latin for "of doves" from "columba", "dove

Figure 6: Hard Negative Cases of Multi-Modal Retrieval. These hard negatives are top-retrieved multi-modality documents from CLIP-DPR.

For both cases, the hard negatives from different modalities showcase some necessary semantics, which are needed by the retrieval model to find relevant information. For example, text documents in Case 1 can provide background knowledge of the Olympic Games and Centennial Olympic Park; image documents in Case 2 supply the "doves" semantics from the visual modality. These informative documents from different modalities can provide sufficient clues to guild dense retrievers to learn necessary semantics during contrastive training.

