# OpenReview forum: "Universal Vision-Language Dense Retrieval: Learning A Unified Representation Space for Multi-Modal Retrieval"
_ICLR.cc/2023/Conference — ICLR 2023 poster_

### Official Review · Reviewer_vCSr · 2022-10-22

**Confidence:** 4
**Correctness:** 2
**Technical Novelty And Significance:** 2
**Empirical Novelty And Significance:** 3
**Recommendation:** 6

**Clarity, Quality, Novelty And Reproducibility:**

The structure of this paper is complete, but more details of the Image verbalization method need to be provided, and the rationality of the Universal embedding optimization strategy needs to be more fully verified.

**Strength And Weaknesses:**

The main strengths of this paper can be concluded as follows:

1. This paper proposes a unified model for multi-modal retrieval, which demonstrates that universal multi-modal search is feasible compared with the divide-and-conquer pipeline with a united model, and also benefits cross-modality tasks.

2. The proposed method bridges the modality gap by optimizing the vision-language embedding space using hard negatives, and aligning the semantics of image captions and figure pixels, which achieves state-of-the-art performance on WebQA datasets.

The main weaknesses of this paper are as follows:

1. There is still a gap between the proposed method and SOTA method in some evaluation tasks on the WebQA Dataset. The authors should provide more corresponding analyses.

2. An important assumption in this paper is that the universal embedding optimization strategy can enable the optimization of the universal embedding space with modality-balanced hard negatives. The authors should provide a mathematical explanation to make the assumption more convincing.

3. Relevant works such as [a] [b] should be compared to make this paper more comprehensive.

[a] Effective Conditioned and Composed Image Retrieval Combining CLIP-Based Features. CVPR2022.
[b] Visual–Textual Hybrid Sequence Matching for Joint Reasoning. TCVB2021.

**Summary Of The Paper:**

This paper proposes a unified model for multi-modal retrieval. The proposed method consists of two techniques, the universal embedding optimization strategy for contrastively optimizing the embedding space, and the Image verbalization method for bridging the modality gap.

**Summary Of The Review:**

This paper is marginally above the acceptance threshold of ICLR2023.

=======================================================================================================

I have checked the comments of the other reviewers and the feedback from the authors carefully, and I would like to keep my score. The authors have answered my concerns, especially the mathematical explanation of the universal embedding optimization strategy.

---

> ### Author Response · Authors · 2022-11-18
> **Reply to Reviewer vCSr**
>
>
> **Reply to weakness 1: There is still a gap between the proposed method and SOTA method in some evaluation tasks on the WebQA Dataset. The authors should provide more corresponding analyses.**
>
> As discussed in our General Response 2, most models focus on the restricted reranking setting, which reranks the candidates that are retrieved by commercial search engines, such as Bing Image Search [1]. It is unfair to compare our retrieval system with the commercial search engine.
>
> In addition, the models developed for reranking focus on a later stage in the search pipeline. The lesser efficiency constraints allow these models to employ a cross encoder architecture to establish interactions between queries and documents using Transformer layers. The cross encoder based retriever can not be applied in the fully retrieval setting because of the efficiency requirement in first stage retrieval.
>
> **Reply to weakness 2: An important assumption in this paper is that the universal embedding optimization strategy can enable the optimization of the universal embedding space with modality-balanced hard negatives. The authors should provide a mathematical explanation to make the assumption more convincing.**
>
> Thanks for your suggestion. We have shown the mathematical explanation in Sec. 4.2.
>
> The imbalanced negative sampling method often makes dense retrievers overfit the ranking signals of text-image/text matching. As shown in Figure 4, when we only use image documents or text documents as negatives, the queries are simply assigned far away from the image documents or text documents. It leads to unnecessary modality discrimination during retrieval.
>
> **Reply to weakness 3: Relevant works such as [a] [b] should be compared to make this paper more comprehensive.**
>
> Thanks for your insightful advice. We have cited and discussed them in Sec.1 and Sec.4.
>
> **References**
>
> [1] Yingshan Chang, Mridu Narang, Hisami Suzuki, Guihong Cao, Jianfeng Gao, and Yonatan Bisk. 2022. Webqa: Multihop and multimodal qa. In Proceedings of CVPR.

---

### Official Review · Reviewer_qnyp · 2022-10-22

**Confidence:** 3
**Correctness:** 3
**Technical Novelty And Significance:** 2
**Empirical Novelty And Significance:** 1
**Recommendation:** 5

**Clarity, Quality, Novelty And Reproducibility:**

The paper is reasonably clear. It is well written in good English and the reported results are encouraging. The authors have also provided the codes in the supplementary material which I expect to be useful for it reproducibility. However, I haven't found much originality in the core work, which I commented in the weaknesses section.

**Strength And Weaknesses:**

**Strengths**

(1) The paper addresses a very interesting tasks of visual language research.
(2) The paper is reasonably well presented and written in good English.
(3) Experimental results are quite encouraging.

**Weaknesses**

(1) The novelty is not clear. I was expecting it to be mentioned or specified in the introduction of the paper for better understanding on the contribution of the paper. Furthermore, I am aware of the following existing papers which are also considering multi-modal embedding for different tasks, such as text [1], sketch [4] and both [3,5] based image retrieval, text and image topic modelling [2] etc.

(2) It is not very clear why a simple summation of the representation of image caption and image feature works well for combining two very different types of modalities. I wonder if any other combination (concatenation, outer product etc.) has been ablated or could be interesting to try.

(3) Currently a lot of progress has been made on text, sketch and both based image retrieval which should also be included and discussed within the literature of cross-modal retrieval.

(4) It is not clear why [CLS] and [SEP] tokens special and different.

(5) I think it is worth defining universal representation learning and image verbalization procedure.

(6) In the experimental results table, citation of the baselines and SOTA methods should be given for readability.

[1] Mai et al., Spatial-Semantic Image Search by Visual Feature Synthesis, CVPR, 2017.
[2] Gomez et al., Self-supervised learning of visual features through embedding images into text topic spaces, CVPR, 2017.
[3] Dey et al., Learning Cross-Modal Deep Embeddings for Multi-Object Image Retrieval using Text and Sketch, ICPR, 2018.
[4] Dutta and Akata, Semantically Tied Paired Cycle Consistency for Any-Shot Sketch-based Image Retrieval, IJCV, 2020.
[5] Sankgkloy et al., A Sketch Is Worth a Thousand Words: Image Retrieval with Text and Sketch, ECCV, 2022.

**Summary Of The Paper:**

This paper presents universal vision-language dense retrieval model, which builds a unified model for multi-modal retrieval. The proposed model encodes queries and multi-modal resources in an embedding space for searching candidates from different modalities. To learn a unified embedding space for multi-modal retrieval, this work has come out with (1) universal embedding optimization strategy, which contrastively optimizes the embedding space using the modality-balanced hard negatives; (2) image verbalization method, which bridges the modality gap between images and texts in the raw data space.

**Summary Of The Review:**

Taskwise, the paper is interresting. It is well presented and also the reported results are also interesting. Howevere, I have found the novelty to be limited.

---

> ### Author Response · Authors · 2022-11-18
> **Reply to Reviewer qnyp**
>
> **Reply to weakness1: The novelty of UniVL-DR.**
> We have discussed related models in Sec. 3. The multi-modal retrieval task is different from sketch/text based image retrieval tasks and focuses on these aspects:
> * Relevance modeling between queries and multi-modality documents instead of object matching
> * Single/cross-modality matching
> * Routing queries to different modalities for returning documents from texts, images, or both of them
> * Retrieval result fusion
>
> These sketch/text based image retrieval methods play a similar role as CLIP--we can view CLIP as a recent strong method in this category.
> UniVL-DR performs better than CLIP because of the following contributions:
> * UniVL-DR focuses on learning a universal embedding space for the multi-modal retrieval task with the modality-balanced hard negative training method and image verbalization method.
> * We develop the modality-balanced hard negative training method, which better guides multi-modal retrievers to route queries to different modalities, conduct a more uniform embedding space, and better distinguish related documents.
> * Our image verbalization method better alleviates the modality gap between images and texts.
>
> **Reply to weakness2: Evaluation on the combination methods of image representations.**
> We add experiments in Appendix A.5. The sum operation works the best empirically. It is a simple method and we do not claim its novelty. In fact, summing up embeddings from different signals and training the full network end-to-end is a standard approach with pretrained Transformers. For example, the token embeddings and position embeddings are summed up in BERT.
> * Concatenation: It separates the representations of image captions and features into subspaces, which is hard to learn.
> * Outer Product: Using outer product to combine representations will conduct an orthogonal image embedding to the embeddings of image captions and features. It is not a typical way to combine representations.
> * Sum: It is like what BERT does in combining token and position embeddings.
>
> | Model | MRR@10 | MRR@20 | NDCG@10 | NDCG@20 |
> |--------------------------|:------:|:------:|:-------:|:-------:|
> | CLIP-DPR (Concatenation) | 23.18 | 23.61 | 20.74 | 22.94 |
> | CLIP-DPR (Outer Product) | 39.89 | 40.47 | 37.40 | 40.22 |
> | CLIP-DPR (Sum) | **48.73** | **46.27** | **49.24** | **49.06** |
>
> **Reply to weakness3: Existing work on text, sketch and both based image retrieval.**
> We have discussed these work in Sec. 3. The multi-modal retrieval task is different from text/sketch-based image retrieval tasks, such as MS COCO and Flickr, thus these models can not be directly compared:
> * Existing text/sketch-based image retrieval tasks focus more on matching general objects instead of relevance modeling.
> * Text/sketch-based image retrieval aims to return matched images. But multi-modal retrieval aims to return images, texts, or both of them, without knowing which modality will provide the right information for the given query.
> * The multi-modal retrieval task is more challenging and targets the following problem:
>     * Relevance modeling between queries and documents.
>     * Modality selection.
>     * Retrieval result fusion.
>
> Besides, existing text/sketch-based image retrieval models[1] also continuously fine-tune CLIP to achieve convincing results. Our method significantly outperforms CLIP, which shows that the techniques we proposed are critical for universal multi-modal methods to outperform divide-and-conquer methods.
>
> **Reply to weakness4: It is not clear why [CLS] and [SEP] tokens are special and different.**
> Both [CLS] and [SEP] tokens are general terminology and widely used in Transformer based models, such as BERT and VinVL. These tokens are added to play special roles in the model, which are different from normal text tokens. Specifically, BERT added [CLS] to gather the full sequence representation and [SEP] indicates the sequence boundary.
>
> **Reply to weakness5: I think it is worth defining universal representation learning and image verbalization procedure.**
>
> * *Universal Representation Learning:* It aims to map queries, texts, and images in one embedding space, which helps to do relevance modeling, modality routing, and retrieval result fusion in this space.
> * *Image Verbalization:* As shown in Sec. 4 and Appendix A.2, we come up with two image verbalization methods, image caption generation, and query generation, using mask language modeling.
>     * The image caption generation method directly uses the VinVL-based image caption model, which is trained with MS COCO.
>     * The query generation method feeds image features and captions to VinVL and is continuously trained to generate image-related queries.
>
> We also add these definitions in Sec. 3.
>
> **Reply to weakness6: Cite baselines and SOTA methods.**
> We have added citations in Table 2 and show more details in Appendix A.1.
>
> [1] A Sketch Is Worth a Thousand Words: Image Retrieval with Text and Sketch. In Proceedings of ECCV.

---

### Official Review · Reviewer_FZm3 · 2022-10-24

**Confidence:** 4
**Correctness:** 3
**Technical Novelty And Significance:** 2
**Empirical Novelty And Significance:** 2
**Recommendation:** 6

**Clarity, Quality, Novelty And Reproducibility:**

This paper is well-written. It's clear and easy to follow, but the novelty may be limited.

**Strength And Weaknesses:**

Strength:
Reasonable motivation and a combination approach.

Weakness:
The proposed image verbalization method simply adopts the existing approach to generate matched captions or queries according to pictures. The performance of the combined method may be bounded by the methods of proposal generation and caption generation models. Is there a strong reason the authors choose VinVL (Zhang 2021) and MLM (Devlin 2019)? Or, it can be any arbitrary methods to achieve this.  In addition, what is the definition of hard negatives,  some visualization examples may help? What will it happen if the proposed method uses non-hard negatives compared to other methods? Are those good results from the hard negatives?
Can the authors explain why to choose NDCG@10, instead of @1, @5, as we know that most of the time the top retrieved results are the most important to queries?  When good results on MRR@10 and @20, why BM25 & Clip-DPR Rec@20 is better a lot in Table 2? Does that imply the UniVl-DR + BM25 method can provide even better?

**Summary Of The Paper:**

This paper proposes a universal vision-language dense retrieval with two techniques, using modality-balanced hard negatives for optimization and bridging the modality gap by the image verbalization method.  Experiments are conducted on the built open-domain dataset from WebQA and compared to existing models for single-modality, divide-and-conquer, and multi-modal retrieval and ablation studies.

**Summary Of The Review:**

The paper proposes to build a universal embedding space for single and cross-modality matching for retrieval tasks with a multi-modality training strategy for queries and documents and try to mitigate the modality boundary between vision and language. I have some concerns (see weakness). If my concerns are addressed, I will consider increasing the score appropriately.

---

> ### Author Response · Authors · 2022-11-18
> **Reply to Reviewer FZm3**
>
> We thank you for the insightful advice. Our paper proposes UniVL-DR and focuses on building a universal embedding space for multi-modal retrieval. UniVL-DR uses a modality-balanced hard negative sampling method and image verbalization method to break the modality gap, effectively select modality, and conduct better multi-modality retrieval results.
>
> **Q1: The reasons for choosing Vision-Language Pretriained (VLP) Models and Pretrained Language Models (PLM).**
>
> * UniVL-DR can use attribute VLP models and PLM models as text/image encoders.
> * We employ VinVL [1] and CLIP [2] in our experiments because they are two representative vision-language models. VinVL is an MLM based model, which is a typical cross-encoder architecture. CLIP encodes images and texts respectively and conducts cross-modality matching using cosine similarity, which is a dual-encoder architecture [3].
> * MLM is one of the most effective pretraining methods for both language modeling and vision-language modeling.
>
> **Q2: The definition of hard negatives, some visualization examples may help? What will it happen if the proposed method uses non-hard negatives compared to other methods?**
>
> * The hard negatives are top-ranked images and texts from the CLIP-DPR retriever. We show some hard negative cases in Appendix A.7.
> * As shown in Table 4, hard negatives are crucial and can significantly improve the retrieval effectiveness more than in-batch negatives.
> * These non-hard negatives are usually uninformative to optimize the dense retrievers, which leads to small training loss and yields diminishing gradient norms [4].
>
>
> **Q3: Are those good results from the hard negatives?**
>
> As shown in Table 4, only using naive hard negative sampling as in previous research [4] shows less effectiveness in improving multi-modal retrieval effectiveness. We need modality-balanced hard negative sampling during training multi-modal dense retrievers.
>
> **Q4: The reason why to choose NDCG@10, instead of @1, @5.**
>
> * In our experiments, we follow widely used retrieval benchmarks, BEIR [5] and MS MARCO [6], and use the evaluation metrics in these papers. All evaluation codes are from pytrec_eval and MS MARCO.
> * We also report some additional experiment results using MRR@1 and NDCG@5 in Appendix A.6. These metrics also share a similar trend and the effectiveness of UniVL-DR stays with different depths of evaluation metrics.
>
> | Model | MRR@1 | NDCG@5 |
> |-----------------------------------|:-----:|:------:|
> | BM25 | 41.14 | 46.24 |
> | BM25 & CLIP-DPR | 7.43 | 19.47 |
> | BM25 & CLIP-DPR (Oracle Modality) | 46.19 | 54.20 |
> | NQ-ANCE | 34.21 | 39.16 |
> | CLIP-DPR | 34.92 | 42.30 |
> | UniVL-DR | **47.56** | **55.28** |
>
> **Q5: When good results on MRR@10 and @20, why BM25 & Clip-DPR Rec@20 is better a lot in Table 2? Does that imply the UniVl-DR + BM25 method can provide even better?**
>
> * BM25 & CLIP-DPR (Oracle Modality) is a "cheating" setting. It uses the testing label to choose the correct modality during retrieval. We show the results of this method for reference, e.g., display the gap between our UniVL-DR and oracle methods.
> * As shown in Table 2 and Table 8, UniVL-DR performs quite close to the oracle in finding the right modality. On the contrary, without the oracle modality information, BM25 & CLIP-DPR performs much worse than UniVL-DR.
>
> We will improve the description to avoid this confusion in our next version.
>
> **References**
>
> [1] Pengchuan Zhang, Xiujun Li, Xiaowei Hu, Jianwei Yang, Lei Zhang, Lijuan Wang, Yejin Choi, and Jianfeng Gao. 2021. Vinvl: Revisiting visual representations in vision-language models. In Proceedings of the IEEE/CVF Conference on Computer Vision and Pattern Recognition.
>
> [2] Alec Radford, Jong Wook Kim, Chris Hallacy, Aditya Ramesh, Gabriel Goh, Sandhini Agarwal, Girish Sastry, Amanda Askell, Pamela Mishkin, Jack Clark, Gretchen Krueger, and Ilya Sutskever. 2021. Learning transferable visual models from natural language supervision. In Proceedings of ICML, pages 8748–8763.
>
> [3] Vladimir Karpukhin, Barlas Oguz, Sewon Min, Patrick Lewis, Ledell Wu, Sergey Edunov, Danqi Chen, and Wen-tau Yih. 2020. Dense passage retrieval for open-domain question answering. In Proceedings of EMNLP.
>
> [4] Lee Xiong, Chenyan Xiong, Ye Li, Kwok-Fung Tang, Jialin Liu, Paul N. Bennett, Junaid Ahmed, and Arnold Overwijk. 2021. Approximate nearest neighbor negative contrastive learning for dense text retrieval. In Proceedings of ICLR.
>
> [5] Nandan Thakur, Nils Reimers, Andreas Rücklé, Abhishek Srivastava, and Iryna Gurevych. 2021. BEIR: A Heterogenous Benchmark for Zero-shot Evaluation of Information Retrieval Models. In Proceedings of NeurIPS.
>
> [6] Payal Bajaj, Daniel Campos, Nick Craswell, Li Deng, Jianfeng Gao, Xiaodong Liu, Rangan Majumder, Andrew McNamara, Bhaskar Mitra, Tri Nguyen, et al. 2016. Ms marco: A human generated machine reading comprehension dataset.

---

### Author Response · Authors · 2022-11-18
**General Response**

We thank all reviewers for their efforts and valuable suggestions. Here we try to address the common concerns.

**1. The updates of our revised paper.**

* *Sec. 4.2:* We add the mathematical explanation to show the motivations of our modality-balanced hard negative training method.
* *Table 2:* We cite related work in our experimental results.
* *Sec. 3:* We cite more related work and discuss more the differences between multi-modal retrieval tasks and text/sketch-based image retrieval tasks. We claim the definitions of universal multi-modal retrieval and image verbalization methods more clear.
* *Sec.1 and Sec.3:* We cite more related work to better introduce our motivations.
* *Appendix A.2:* We describe the details of image verbalization methods in more detail.
* *Appendix A.5:* We add experiments to evaluate different methods by combining the representations of image captions and image features.
* *Appendix A.6:* We add experiments to show higher-ranked retrieval results using additional evaluation metrics, NDCG@5 and MRR@1.
* *Appendix A.7:* We add some hard negative cases.

**2. SOTA retrieval results on the WebQA dataset.**

* WebQA [9] conducts two evaluation settings: full retrieval and restricted reranking.
* The restricted reranking setting aims to rerank candidate documents that are retrieved by commercial search engines, such as Bing Image Search [12]. The commercial search's retrieved results, as expected, are pretty good. It is unfair to compare academic methods, which focus on one novel idea, with a commercial system that leverages years of engineering and research effort, as well as being trained on much more data.
* The fully retrieval baseline [9] employs BM25 and CLIP (Zero-Shot) as image and text retrievers. In our experiments, we use them to conduct a stronger baseline model, CLIP-DPR & BM25 (Oracle Modality). It employs CLIP-DPR and BM25 as retrievers and additional uses the test label to select modality during retrieval. Our evaluation shows that our method is better than BM25 and CLIIP (Zero-Shot) and not far away from CLIP-DPR & BM25 (Oracle Modality).

**3. The differences between multi-modal retrieval tasks with retrieval tasks.**

We summarize the characteristics of different retrieval tasks in the following table. As we can see, the multi-modal retrieval task targets relevance modeling, modality selection, ranking result fusion, and cross-modality matching, which is different from other image/text retrieval tasks.

* Matching Type: The types of matching between queries and documents.
    * *Relevance* indicates searching related documents for queries, for example, to find relevant answers to questions, which require new information.
    * *Paraphrase* means the texts or images share the same semantics. It is more about aligning the modality rather than answering a question.
* Modality selection: Selecting the candidate documents from which modality.
* Ranking Result Fusion: Fusing multi-modality documents as the retrieval result.
* Cross modality matching: Retrieving documents from a different modality.

| Tasks | Related Work | Matching Type | Modality Selection | Ranking Result Fusion | Cross Modality Matching |
|------------------------------|:------------:|:------------------:|:-----------------:|:---------------------:|:-----------------------:|
| Text-Text Retrieval | [1,2,3] | Relevance | | | |
| Text-Image Retrieval | [4,5,6,7,8] | Paraphrase | | | ✓ |
| Image-Text Retrieval | [4,5,6,7] |  Paraphrase | | | ✓ |
| Image/Sketch-Image Retrieval | [10,11] | Paraphrase | | | |
| Multi-Modal Retrieval | [9] | Relevance | ✓ | ✓ | ✓ |

**References**

[1] BEIR: A Heterogenous Benchmark for Zero-shot Evaluation of Information Retrieval Models. In Proceedings of NeurIPS.

[2] Ms marco: A human generated machine reading comprehension dataset.

[3] Dense passage retrieval for open-domain question answering. In Proceedings of EMNLP.

[4] Microsoft coco captions: Data collection and evaluation server.

[5] From image descriptions to visual denotations: New similarity metrics for semantic inference over event descriptions. Transactions of the Association for Computational Linguistics.

[6] Vinvl: Revisiting visual representations in vision-language models. In Proceedings of CVPR.

[7] Learning transferable visual models from natural language supervision. In Proceedings of ICML.

[8] Learning cross-modal deep embeddings for multi-object image retrieval using text and sketch. In 24th International Conference on Pattern Recognition.

[9] Webqa: Multihop and multimodal qa. In Proceedings of CVPR.

[10] Image-to-Image Retrieval by Learning Similarity between Scene Graphs. In Proceedings of AAAI.

[11] Semantically Tied Paired Cycle Consistency for Zero-Shot Sketch-Based Image Retrieval. In Proceedings of CVPR.

[12] WebQA: A Multimodal Multihop NeurIPS Challenge. In Proceedings of the NeurIPS 2021 Competitions and Demonstrations Track.

---

### Decision · Program_Chairs · 2023-01-20

**Decision:**

Accept: poster

**Justification For Why Not Higher Score:**

already high

**Justification For Why Not Lower Score:**

NA

**Metareview: Summary, Strengths And Weaknesses:**

This paper got mixed reviews. The reviewers generally find the task interesting although some were not convinced about its novelty. I am in favor of accepting the paper as it has some interesting findings.

**Note From Pc:**

if the above contains the word "oral" or "spotlight" please see: "oral" presentation means -> notable-top-5% and "spotlight" means -> notable-top-25%. As stated in our emails, we are disassociating presentation type from AC recommendations

**Summary Of Ac-Reviewer Meeting:**

NA